# Nutraceutical Screening in a Zebrafish Model of Muscular Dystrophy: Gingerol as a Possible Food Aid

**DOI:** 10.3390/nu13030998

**Published:** 2021-03-19

**Authors:** Rosario Licitra, Maria Marchese, Letizia Brogi, Baldassare Fronte, Letizia Pitto, Filippo M. Santorelli

**Affiliations:** 1Molecular Medicine & Neurobiology—ZebraLab, IRCCS Fondazione Stella Maris, 56128 Pisa, Italy; rosario.licitra@fsm.unipi.it (R.L.); maria.marchese@fsm.unipi.it (M.M.); 2Department of Veterinary Science, University of Pisa, 56124 Pisa, Italy; letizia.brogi@sns.it (L.B.); baldassare.fronte@unipi.it (B.F.); 3Bio@SNS, Department of Neurosciences, Scuola Normale Superiore, 56126 Pisa, Italy; 4Istituto Fisiologia Clinica IFC-CNR, 56124 Pisa, Italy; l.pitto@ifc.cnr.it

**Keywords:** nutraceuticals, nutrition, Duchenne muscular dystrophy, zebrafish, gingerol, heme oxygenase 1, mitochondrial respiration, locomotion

## Abstract

Duchenne muscular dystrophy (DMD), caused by mutations in the dystrophin gene, is an inherited neuromuscular disorder that causes loss of muscle mass and motor skills. In the era of genomic medicine, there is still no known cure for DMD. In clinical practice, there is a growing awareness of the possible importance of nutrition in neuromuscular diseases. This is mostly the result of patients’ or caregivers’ empirical reports of how active substances derived from food have led to improved muscle strength and, thus, better quality of life. In this report, we investigate several nutraceutical principles in the *sapje* strain of zebrafish, a validated model of DMD, in order to identify possible natural products that, if supplemented in the diet, might improve the quality of life of DMD patients. Gingerol, a constituent of fresh ginger, statistically increased the locomotion of mutant larvae and upregulated the expression of heme oxygenase 1, a target gene for therapy aimed at improving dystrophic symptoms. Although three other compounds showed a partial positive effect on locomotor and muscle structure phenotypes, our nutraceutical screening study lent preliminary support to the efficacy and safety only of gingerol. Gingerol could easily be proposed as a dietary supplement in DMD.

## 1. Introduction

Duchenne muscular dystrophy (DMD) is an inherited genetic disease caused by mutations in the DMD gene, leading to muscle fiber degeneration and loss of dystrophin in skeletal muscle. The disease affects males and occurs in around 1 in 3500 boys. The symptoms appear in early infancy and include progressive muscle weakness and cardiac and respiratory muscle deficiencies. As the disease progresses, muscle tissue shows a strong inflammatory reaction, becomes atrophied, and is ultimately replaced with fat and fibrotic tissue. This process results in loss of ambulation, impaired lung and heart function, and reduced life expectancy [1]. Steroid therapy and other supportive strategies have improved survival and health-related quality of life in this setting, but steroid treatment has several short-term side effects and, in the long-term, does not prevent loss of autonomous ambulation. More recently, research has focused on possible gene-based strategies, and a few of these are now reaching the phase II/III clinical trial stage. They include, among others, upregulation of utrophin, enhancement of muscle regeneration, and virus-mediated mini-dystrophin gene therapy (www.clinicaltrials.gov; NCT03362502—accessed on 1 February 2021) [2]. Moreover, recently, the use of a group of cells called muscle side population (SP) cells to deliver genes, such as the human microdystrophin, showed that these SP cells are capable of recapitulating the myogenic lineage. Therefore, this approach may have substantial implications for cell-based therapy for Duchenne muscular dystrophy [3,4]. Meanwhile, two therapies have been approved by international regulatory agencies: one is based on stop codon readthrough, whereas the other uses an exon-skipping strategy to restore production of internally deleted, yet efficient, dystrophin [5]. Nonetheless, approved drugs have shown only partial success in DMD and, moreover, remain suitable for relatively few patients, i.e., those with specific mutations.

In real-world clinical practice, DMD patients seen in neuromuscular disease centers often report that a diet supplemented with bioactive compounds improves their muscle strength [6]. Recently, there has been growing recognition, stemming from empirical observations in DMD patients and studies in the mdx mouse model of the disease [7], that proper nutrition can exert anti-inflammatory effects and slow the rate of muscle atrophy. For instance, resveratrol, a natural nutraceutical produced by plants, likely improves the dystrophic myopathology present in human cells and animal models [8]. Although these observations have opened up a new area of research into chronic muscle diseases [9,10,11,12], similar studies in larger experimental models of DMD are needed to ascertain the potential efficacy, and safety, of nutraceuticals in vivo.

The zebrafish (*Danio rerio*) is a popular vertebrate for modeling neuromuscular disorders and exploring possible new drug treatments. For research purposes, this species has a number of advantages over mice: fecundity, genetic homology, transparency, and inexpensiveness. The use of zebrafish for drug discovery will not replace in full for mammalian assays, however the identification of the scale and efficiency of novel compounds and molecules is important for a perfect replication of mouse and human physiology. Thus, zebrafish are useful to speed up the identification of new molecules for phenotype-based discovery leading to preliminary assessments of compounds in vivo and bridging the gap between in vitro models and mammalian models [13].

The *sapje* strain, a validated model of DMD [14], carries a recessive nonsense mutation in dystrophin and exhibits muscle disorganization, motor dysfunction, and early death. The *sapje* zebrafish phenotype appears at 3 days post fertilization (dpf), and its severity recalls that of DMD in children. Successful high-throughput screenings using *sapje* [14] have confirmed the value of the species for developing new pre-clinical hypotheses. Fluoxetine, sildenafil, and dasatinib are effective compounds validated in *sapje* and now in the pipeline for randomized clinical trials [15,16,17]. Nonetheless, drugs emerging from recent screenings show several side effects and, although already approved for use in humans, cannot be routinely proposed in clinical settings until further, more costly, preclinical investigations have been performed.

We hypothesized that a nutraceutical screening study targeting early dysfunctional pathways underlying DMD might identify compounds capable not only of improving clinical phenotypes, but also of being rapidly and safely applied in clinical settings through dietary supplements or patient lifestyle modifications. 

## 2. Materials and Methods 

### 2.1. Zebrafish Care and Maintenance

Adult male and female wild-type AB and *sapje* zebrafish strains (from the lab of S.M. Hughes) [18] were maintained according to standard procedures [19] on a 14 h light:10 h dark cycle. In our experimental setting, we did not select a specific sex. Indeed, fish larvae sex was mixed, since the first indication of zebrafish sex differentiation occurs at 10–12 dpf, when the ovarian gonocyte proliferates and differentiates [20]. Handling of zebrafish embryos and larvae complied with the guidelines of our internal institutional animal care committee, and experiments were performed in accordance with, and under the supervision of, the Institutional Animal Care and Use Committee of the University of Pisa. Every effort was made to minimize both animal suffering and the number of animals needed to collect reliable scientific data. 

### 2.2. Genotyping Sapje

Genomic DNA extracted from fish was used as the PCR template and sequenced in order to identify the specific mutation in the dystrophin gene of *sapje*. The primers and PCR conditions used have already been described elsewhere [14].

### 2.3. Drugs

We tested 19 natural compounds listed in the Enzo Natural Products Library (Enzo Life Sciences, Farmingdale, NY, USA), plus another 4 compounds from Sigma (Sigma-Aldrich, St. Louis, MO, USA). The selected 23 molecules have been chosen on the basis of their anti-inflammatory or antioxidant actions on muscle tissue. All compounds were added to the embryo medium at 24 hours post fertilization (hpf), some at different concentrations. 

Compounds were added to plates with concentrations ranging from 1 to 100 μM with the average concentration being 10 μM, following the range suggested for zebrafish drug screenings [21]. Table 1 lists the nutraceutical principles and concentrations used and their effects on *sapje* larvae (*n* = 40 for each experimental group). The endpoint parameters used for this evaluation were mortality, locomotion, and muscle structure.

### 2.4. Birefringence Analysis

At 96 hpf, the *sapje* larvae were analyzed using the birefringence method to identify and count phenotypically dystrophic (homozygous) and non-dystrophic (heterozygous) specimens, as described by Kawahara et colleagues [14]. The analysis was performed on larvae anesthetized using Tricaine-MS-222 (Sigma-Aldrich, St. Louis, MO, USA); each larva was placed between two overlapping polarized lenses, and the muscle structure was observed under a Leica M205FA stereo-microscope (Leica, Wetzlar, Germany), using magnification ranging from 7.5× to 60×. Briefly, the trunk muscle striation was examined using the incident light of the microscope, turning the top polarizing lens until the light refracting through the striated muscle was visible. On the basis of the findings, the tested larvae were classified as dystrophic or non-dystrophic.

### 2.5. Locomotion Analysis

Locomotion behavior (distance and velocity) was measured in untreated and treated homozygous *sapje* larvae at 5 dpf using the Daniovision system (a complete activity monitoring system for zebrafish larvae) connected with a specific video tracking software: the Ethovision XT12 (Noldus Information Technology, Wageningen, The Netherlands). Briefly, single larvae were taken from the rearing dishes and transferred into 96 multiwell plates containing 100 μL egg water per well. Then, one plate at a time was placed in the DanioVision system, and larval locomotor activity was monitored for 30 min (Appendix A).

### 2.6. RNA Isolation and Real-Time PCR Analysis

Total RNA was extracted from untreated and treated homozygous *sapje* larvae at 5 dpf using the Quick RNA Miniprep Kit (Zymo Research, Irvine, CA, USA). cDNA was synthesized by reverse transcription of about 1 μg of total RNA, and quantitative real-time polymerase chain reaction (qRT-PCR) was performed in qPCRBIO SyGreen Mix Hi-ROX (PCR Biosystem, Wayne, NJ, USA) using the following primers: forward primer for zebrafish heme oxygenase 1 (hmox1F) cDNA (NM_001127516): 5′ ATGGACTCCACCAAAAGC 3′; reverse primer for zebrafish heme oxygenase 1 (hmox1R) cDNA: 5′ CCTTTCTGGTAGCTGAGCATC 3′; forward primer for zebrafish elongation factor 1 alpha (EF1 alphaF): 5′ CTGGAGGCCAGCTCAAACAT 3′, reverse primer for zebrafish elongation factor 1 alpha (EF1 alphaR): 5′ ATCAAGAAGAGTAGTACCGCTAGCATTAC 3′, as described in [13], and the Mic Real-Time PCR System (Bio Molecular Systems, Upper Coomera, Australia). Relative mRNA expression was quantified using the comparative Ct (ΔCt) method and expressed as 2^−ΔΔCt^. All data were normalized to EF1 alpha as a loading control and using expression in the adult stage as the calibrator. Each assay was done in triplicate, and we studied 75 larvae per group.

### 2.7. Mitochondrial Respiratory Analysis

Mitochondrial respiration was analyzed in untreated and treated homozygous *sapje* larvae at 5 dpf using the XF24 extracellular flow analyzer (Seahorse Bioscience, Billerica, MA, USA). The dual analyte sensor cartridges were soaked in XF calibrator solution (Seahorse Bioscience, Billerica, MA, USA) in 24-cell culture microplates (Seahorse Bioscience, Billerica, MA, USA) overnight at 28 °C to hydrate. About 30 min before the trial period, the appropriate injection cartridges were reloaded. The following chemicals were used for this experiment: oligomycin at a concentration of 25 µM, FCCP at a concentration of 5 µM, and rotenone plus antimycin A at a concentration of 5 µM. The 5 dpf larvae were staged and placed in 20 of the 24 wells of an islet microplate. The islet plate acquisition screens were placed on the measurement area to hold the larvae in place. Four wells were left empty as a control. Each well was filled with 500 µL of egg water (pH 7.4). Basal respiration, ATP production, maximal respiration rate, and spare respiratory capacity were measured using a standard approach [22].

### 2.8. Statistical Analysis

All data were analyzed applying either parametric or non-parametric methods, depending on the distribution of the response variable in question, shown by the Shapiro-Wilks test. Homogeneity of variance was assessed using the Levene test. Post hoc comparisons were performed using the Mann-Whitney test with Bonferroni’s correction or an unpaired t test following non-parametric analysis of variance. All statistical analysis was performed using GraphPad Prism (GraphPad Software, Inc., CA, USA).

## 3. Results

### 3.1. Analysis of Birefringence and Locomotion of Treated Sapje

We tested a total of 23 molecules, some at more than one concentration (see Table 1). Several compounds showed a negative outcome for mortality, birefringence, and locomotion assessments. These included idebenone, a drug recently suggested (in a phase III trial) to show efficacy on respiratory function in DMD patients. Conversely, four molecules—gingerol, resveratrol, flavokawain A, and pterostilbene—appeared promising, as they have allowed a partial rescue at concentrations equal to 4, 5, 4 and 2 µM, respectively (Figure 1). Moreover, compared with untreated *sapje* larvae, the ones treated with resveratrol showed, on average, greater trend in locomotion (1030 vs. 546 mm for distance and 0.53 vs. 0.28 mm/s for velocity) and a lower rate of impaired muscle structure on birefringence assay (11 vs. 22%). However, these differences were not statistically significant. Administration of gingerol (4 µM), on the other hand, significantly modified locomotion (*p* ≤ 0.05), while it showed no effect on muscle structure.

In order to strengthen the statistical data on the effect of gingerol on locomotion in *sapje* larvae, we performed an additional trial in homozygous larvae, increasing the number of analyzed fish (Appendix A). The results of this second locomotion test confirmed that gingerol increased both distance and velocity displayed by treated (*n* = 128) compared with untreated *sapje* (*n* = 103) (*p* ≤ 0.0001).

### 3.2. Effect of Gingerol on the Hmox1 Pathway

Since hmox1 can be considered a novel target for obtaining improvement of dystrophic symptoms [16], a qRT-PCR analysis was also performed in the current study. The hmox1 mRNA expression findings suggested that gingerol impacts the hmox1 pathway by increasing its expression (Figure 2). In particular, gingerol-treated homozygous *sapje* larvae showed more than twice the hmox1 expression shown by untreated ones.

### 3.3. Effect of Gingerol on Mitochondrial Function

In view of a recent study highlighting the effect of gingerol on mitochondrial biogenesis and function [23], we evaluated mitochondrial respiration in dystrophic larvae treated with 4 µM gingerol versus untreated siblings. Micro-oxygraphy analysis showed no significant differences between the two experimental groups (Appendix A). 

## 4. Discussion

In this study, we investigate the effect of several nutraceutical principles in the *sapje* strain, a validated zebrafish model of DMD, in order to identify possible natural products that, if supplemented in the diet, might improve the quality of life of DMD patients. We found that Gingerol, a constituent of fresh ginger, statistically increased the locomotion of mutant larvae and upregulated the expression of heme oxygenase 1, a target gene for therapy aimed at improving dystrophic symptoms. The search for treatment protocols able to reduce the severity and progression of muscle wasting in DMD is ongoing. Even though successful treatments have been identified, they are burdened by side effects or applicable only in a limited number of patients [24]. Increasing evidence shows that oxidative stress is involved in the pathogenesis of DMD: loss of sarcolemmal integrity due to absence of dystrophin leads to a cascade of pathological events [25]. Indeed, oxidative stress results from an imbalance in the production of reactive oxygen species (ROS) and their removal by specific defense systems, namely antioxidants. Unless the ROS are removed by antioxidants, ROS accumulation occurs, which ultimately leads to cell death and tissue degeneration. Therefore, the use of molecules with an antioxidant property can act either directly, by scavenging free radicals, or indirectly, by increasing exogenous cellular defenses. In fact, nutraceuticals with antioxidant capabilities could be beneficial in DMD; therefore, there are some antioxidant nutraceuticals trialed in DMD including Coenzyme Q10, melatonin, and preparations of traditional Chinese medicine [26]. Recently, antioxidant drugs administered to *mdx* mice, *sapje* zebrafish, and DMD patients have shown potential benefits, improving several parameters used to assess muscular dystrophy, such as muscle structure and function, locomotor activity, and respiratory function [27,28,29].

In this study, 23 molecules were selected for a nutraceutical screening either on the basis of their role as phosphodiesterase inhibitors, which have a known anti-inflammatory action on muscle, or because they are antioxidants known to counteract enhanced free radical production [30]. Mortality, improved locomotion, and a reduced number of larvae with impaired birefringence were taken as efficacy endpoints [31].

Most of the compounds screened did not significantly improve or modify these two endpoints. However, resveratrol, flavokawain A, and pterostilbene have allowed a limited rescue of the muscle structure and locomotor phenotype. Two of these four compounds, because of their structural similarities to resveratrol [32], have shown anti-cancer activities, i.e., inducing apoptosis (flavokawain A) [33], or displaying antioxidant, anti-inflammatory, and anti-carcinogenic properties (pterostilbene). Resveratrol, a natural polyphenolic compound produced by plants, is also reported to exhibit antioxidant, anti-inflammatory, and anti-proliferative properties in mammalian cells and animal models and might, therefore, exert pleiotropic beneficial effects in different pathophysiological states [34,35]. Grape skin is rich in resveratrol, which can, therefore, be supplemented through diet. In the present study, the best results were obtained with gingerol, the major pharmacologically active component of ginger (*Zingiber officinale*), which significantly increased locomotion in *sapje* larvae versus their untreated siblings (*p* < 0.0001); however, it does not ameliorate the muscle structural damage. The reason for the efficacy of gingerol in the zebrafish model of DMD is unclear but could be related to its known biological (including anti-inflammatory and anti-oxidant) activities [36]. Moreover, gingerol is also a selective inhibitor of subtype B of the phosphodiesterase-4 family of enzymes, which are involved in regulating the release of anti-inflammatory and pro-inflammatory cytokines within cells [37]. Previous drug-screening studies conducted in *sapje* zebrafish showed that sildenafil, an inhibitor of phosphodiesterase-5, whose mechanism of action is believed to affect the Hmox1 pathway and increase Hmox1 protein expression, could improve locomotion and survival [16] in *sapje* larvae. Therefore, it is possible that compounds (both natural and synthetic) acting on the Hmox1 pathway may have positive effects in DMD. Hmox1 might indeed be a novel target for the improvement of dystrophic symptoms [16], as it has also been reported to be downregulated in younger DMD patients [25] and in the *mdx* mouse model [38]. In addition, Hmox1 influences satellite cells and disease progression in mice [39].

## 5. Conclusions

All the above findings suggested that modulation of Hmox1 expression could have beneficial effects on the skeletal muscle phenotype of DMD. Moreover, in vitro study of the effect of 6-gingerol treatment on the expression of antioxidant enzymes showed an up-regulation of the mRNA and protein levels of Hmox1 and, thus, suggested that it might also have preventive and therapeutic potential for the management of neurodegenerative disorders such as Alzheimer’s disease [40]. Our qPCR data showing increased expression of hmox1 mRNA in homozygous *sapje* treated with gingerol, considered in the light of the above findings, seem to show that this molecule exerts its protective effect, at least in zebrafish, by modulating Hmox1, a critical pro-inflammatory molecular marker in DMD. The absence of a significant modulation of oxidative metabolism in response to treatment with this nutrient means that this effect is less likely to be due to free radical production, although we did not test oxidative stress directly in the fish. Overall, data in zebrafish are “first level” information before considering additional, more costly work in mammalian systems. Further insights, from replication of these experiments in larger animal models of DMD, are needed to clarify the role of gingerol at molecular level in cellular signaling cascades [41]. This must, therefore, be the next, crucial step prior to future placebo-controlled studies in patients. In translational terms, although three other compounds showed a partial positive effect on locomotor and muscle structure phenotypes, our nutraceutical screening study lent preliminary support to the efficacy and safety only of gingerol. Gingerol could easily be proposed as a dietary supplement in DMD.

## Figures and Tables

**Figure 1 nutrients-13-00998-f001:**
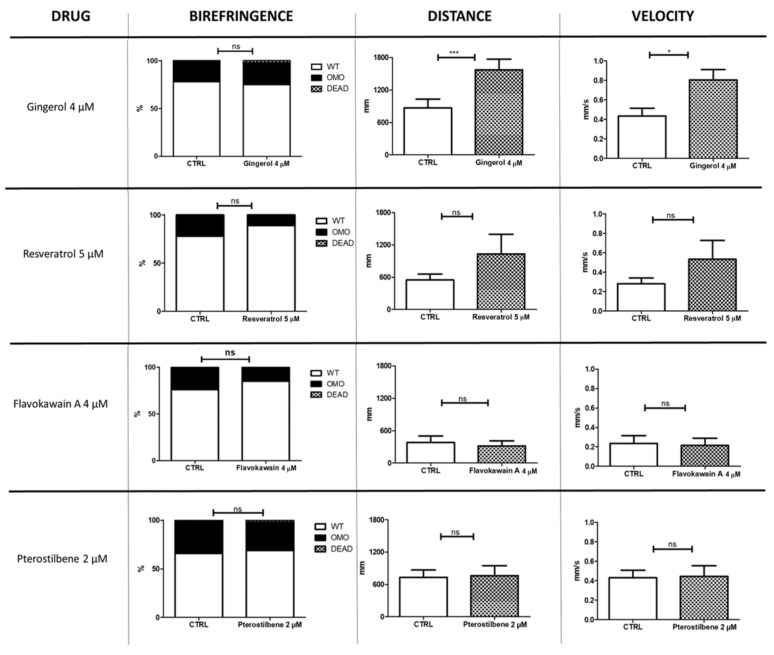
Nutraceutical compounds with positive effects on *sapje* larvae. The figure illustrates the effects of four nutraceutical molecules on muscle structure (birefringence) and locomotion (distance and velocity) of zebrafish larvae at 4 and 5 dpf, respectively (*n* = 40 for each experimental group). The error bars show standard error of the mean (SEM). CTRL, untreated *sapje*; OMO, dystrophic (homozygous) *sapje*; WT, wild-type-like non-dystrophic (heterozygous) *sapje*; DEAD, dead fish, ns *p* > 0.05, * *p* ≤ 0.05, *** *p* ≤ 0.001.

**Figure 2 nutrients-13-00998-f002:**
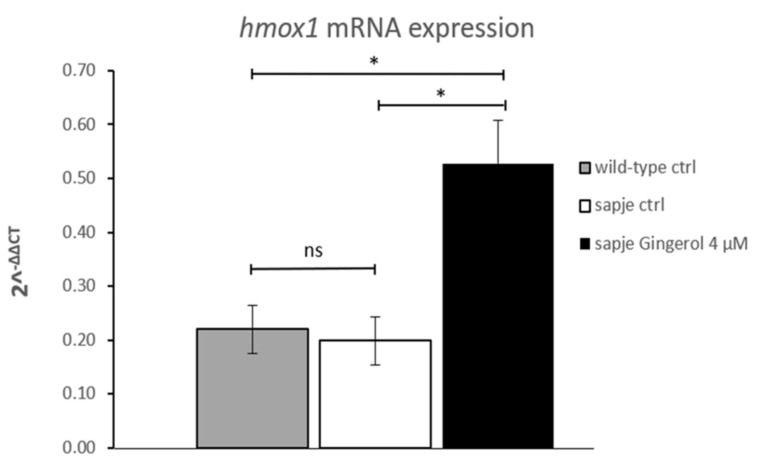
Quantitative real-time PCR analysis of hmox1 mRNA in untreated (wild-type (WT) *n* = 75; *sapje* CTRL, *n* = 75) and in treated (gingerol 4 µM, *n* = 75) homozygous *sapje* larvae. The error bars show the standard error of the mean (SEM). * *p* ≤ 0.05, ns *p* > 0.05.

**Table 1 nutrients-13-00998-t001:** Details of the tested nutraceutical molecules including their effect on *sapje* larvae (*n* = 40 for each treatment).

Bioactive Compound	Concentration (μM)	Effect	Source
Simvastatin	50	Toxic	Sigma
	10	Toxic	
	5	Toxic	
	1	Toxic	
Idebenone	5	Phenotype retained	Sigma
	1	Phenotype retained	
Pterostilbene	10	Toxic	Sigma
	5	Phenotype retained	
	2	Partial rescue	
	1	Phenotype retained	
Resveratrol	1	Partial rescue	Sigma
	0.5	Phenotype retained	
Curcumin	13	Phenotype retained	Natural Products Library
	5	Phenotype retained	
	3	Phenotype retained	
Genistein	9	Phenotype retained	Natural Products Library
Daidzein	9	Phenotype retained	Natural Products Library
Actinomycin D	2	Phenotype retained	Natural Products Library
Gingerol	8	Phenotype retained	Natural Products Library
	4	Partial rescue and increased motility	
Colchicine	6	Phenotype retained	Natural Products Library
Cyclosporin A	2	Toxic	Natural Products Library
Camptotechine	7	Toxic	Natural Products Library
E64D	7	Phenotype retained	Natural Products Library
	4	Phenotype retained	
Gossypol	5	Phenotype retained	Natural Products Library
	2	Phenotype retained	
Epigallocatechin	5	Phenotype retained	Natural Products Library
	3	Phenotype retained	
Quercetin	7	Phenotype retained	Natural Products Library
Resveratrol	11	Phenotype retained	Natural Products Library
	5	Partial rescue	
Shikonin	8	Phenotype retained	Natural Products Library
	4	Phenotype retained	
Kaempferol	8	Phenotype retained	Natural Products Library
	4	Phenotype retained	
Rottlerin	5	Toxic	Natural Products Library
	2	Phenotype retained	
Tomatidine	6	Phenotype retained	Natural Products Library
	3	Phenotype retained	
Ferulic Acid	12	Toxic	Natural Products Library
Flavokawain A	8	Toxic	Natural Products Library
	4	Partial rescue	

## Data Availability

The data that support the findings of this study are available from the corresponding author, upon reasonable request.

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
