# Peer review of "Nutraceutical Screening in a Zebrafish Model of Muscular Dystrophy: Gingerol as a Possible Food Aid"

_nutrients, 2021, doi:10.3390/nu13030998_

Round 1

Reviewer 1 Report

The authors present an interesting investigating using the zebrafish as a drug screening tool. They report that of the 23 compounds tested, 4 appeared promising and that of the 4, gingerol provided the strongest outcome. Furthermore, they show that gingerol increase heme oxygenase 1 which has been shown in many dystrophic studies to be a target gene of the Nrf2 pathway. The manuscript is well written although interpretation of statistics could be improved (see below). The methods appear sound although I was unable to open supplemental data (see below). Data presentation is mostly clear and the discussion is appropriate given the lack of mechanistic investigation.

General comments     

Methods

  • Dystrophy is a predominant male disease with females displaying a reduced pathology. Are the zebrafish all male in this study or a mix?
  • While reading the manuscript I was questioning why or how the 24 compounds were chosen. Choices of nutraceutical was justified in the discussion on line 199-203. This should be in the also made clear in 2.3 Drugs.
  • I was unable to open the supplemental data, so I am assuming that the birefringence methods are described more thoroughly. Following the methods in 2.5, there is no way I could replicate this experiment. It is also recommended that a reference for the methods be included in the main text and not just in the supplemental figures.
  • Its not immediately clear what N was used in the experiments.

Results

  • The interpretation of the statistics is misleading. Initially it is stated that resveratrol, flavokawain and pterostilbene show greater locomotion and lower rate of impaired muscle structure. However, the differences are not significant. No significant difference indicates that results are not different, so the statement saying that these compounds showed greater locomotion and reduced impaired structure should be modified to reflect no differences with these compounds.

Discussion

  • It is my understanding that birefringence is a measure of sarcomere structure and is used in dystrophic zebrafish to show muscle damage. It should be discussed that you saw no change in measures of muscle damage in your gingerol treatment.
  • Line 204 begins with most of the compounds did not significantly However, it seems that gingerol was the only compound that did result in a significant improvement. Additionally, its stated in the next sentence (line 205) that resveratrol, flavokawain and pterostilbene improved locomotion. Yet the statistics say only gingerol was significant. Please modify so the conclusions are a true reflection of the statistical outcomes.

In the discussion lines 226-229 the authors correctly stat that Hmox-1 has been shown to be a potential therapeutic target in dystrophy. However, the interpretation of the references used is unclear and again misleading. Petrillo (#20) and Chan (#32) show that compared to control groups, gene transcription levels of HO-1 is increased and that protein levels were decreased. Figure 2 shows untreated and treated homozygous fish. What are the levels of hmox1 gene in the WT fish?

Reviewer 2 Report

This is an interesting manuscript regarding the role of nutritional substances on ameliorating the symptoms of DMD. There are several major comments that have to be addressed by the authors in order to improve their manuscript.

What is the physiological meaning of the concentrations at which each molecule was tested? Are they realistic for administration in human patients? Please comment this issue extensively in the manuscript and cite relevant literature.

Please explain in more detail the principles of the tests used for the birefringence and locomotion analyses.

According to the table 1, some of the administered substances exerted toxic action. How was this assessed? Which parameters were measured?

Why were the substances added to the embryo medium 24 h post fertilization? Do the authors have any evidence regarding the effects of these substances in case they are administered after the first 3 days of the fish lives, when the sapje zebrafish phenotype appears?

It is suggested that the first paragraph of the discussion describes in brief the main findings of the study.

Please comment how easily the results obtained from experiments in zebrafish can be extrapolated to the mdx mice and to humans.

The authors should add one paragraph in the discussion referring to the role of oxidative stress on DMD onset and to the molecular mechanisms through which several nutritional substances with antioxidant properties could potentially act beneficially.  

It is suggested that the authors cite the following papers to improve their manuscript:

DOI: 10.1073/pnas.0400373101

DOI: 10.1016/j.yexcr.2004.04.010

Author Response

Pease see the attachment

Round 2

Reviewer 1 Report

I thank the authors for addressing all suggestions to the original manuscript. 

Reviewer 2 Report

The authors have fully addressed my comments.